# Comparison of Nutrition Indices for Prognostic Utility in Patients with Sepsis: A Real-World Observational Study

**DOI:** 10.3390/diagnostics13071302

**Published:** 2023-03-30

**Authors:** Django Kyo, Shiho Tokuoka, Shunsuke Katano, Ryo Hisamune, Hidero Yoshimoto, Shuhei Murao, Yutaka Umemura, Akira Takasu, Kazuma Yamakawa

**Affiliations:** 1Faculty of Medicine, Osaka Medical and Pharmaceutical University, 2-7 Daigakumachi, Takatsuki 569-8686, Japan; ompu20191038@s.ompu.ac.jp (D.K.);; 2Department of Emergency and Critical Care Medicine, Osaka Medical and Pharmaceutical University, 2-7 Daigakumachi, Takatsuki 569-8686, Japan; 3Department of Surgery, Osaka Medical and Pharmaceutical University, 2-7 Daigakumachi, Takatsuki 569-8686, Japan; 4Department of Traumatology and Acute Critical Medicine, Osaka University Graduate School of Medicine, 2-15 Yamadaoka, Suita 565-0871, Japan; 5Division of Trauma and Surgical Critical Care, Osaka General Medical Center, 3-1-56 Bandai-Higashi, Sumiyoshi 558-8558, Japan

**Keywords:** sepsis, geriatric nutritional risk index, prognostic nutritional index, controlling nutritional status, prognostic value, poor nutrition

## Abstract

Background: Nutritional status of critically ill patients is an important factor affecting complications and mortality. This study aimed to investigate the impact of three nutritional indices, the Geriatric Nutritional Risk Index (GNRI), Prognostic Nutritional Index (PNI), and Controlling Nutritional Status (CONUT), on mortality in patients with sepsis in Japan. Methods: This retrospective observational study used the Medical Data Vision database containing data from 42 acute-care hospitals in Japan. We extracted data on baseline characteristics on admission. GNRI, PNI, and CONUT scores on admission were also calculated. To evaluate the significance of these three nutritional indices on mortality, we used logistic regression to fit restricted cubic spline models and constructed Kaplan–Meier survival curves. Results: We identified 32,159 patients with sepsis according to the inclusion criteria. Of them, 1804 patients were treated in intensive care units, and 3461 patients were non-survivors. When the GNRI dropped below 100, the risk of mortality rose sharply, as did that when the PNI dropped below about 40. An increased CONUT score was associated with increased mortality in an apparent linear manner. Conclusion: In sepsis management, GNRI and PNI values may potentially be helpful in identifying patients with a high risk of death.

## 1. Introduction

Sepsis and septic shock continue to be major problems in healthcare that affect millions of people worldwide every year [1,2]. Sepsis causes life-threatening organ dysfunction due to an abnormal host response to infection [3]. The vast majority of sepsis occurs in patients in low-income countries, likely due to their poor nutritional condition [4,5].

The nutritional status of critically ill patients in intensive care units (ICUs) rapidly deteriorates, especially during the first week. Malnourished patients may suffer higher rates of complications [6,7], prolonged hospital stays, and poor prognosis [8,9]. Thus, a number of indices have been devised for the assessment of nutritional statuses, such as the Geriatric Nutritional Risk Index (GNRI), the Prognostic Nutritional Index (PNI), and the Controlling Nutritional Status (CONUT) score [10,11,12]. Serum albumin level and body mass index (BMI) are used to calculate the GNRI, whereas the PNI uses serum albumin concentration and total peripheral blood lymphocyte count to assess the systemic immune and nutritional status. Contrastingly, to reflect a patient’s nutrition and immunological status, along with serum albumin concentration and total peripheral lymphocyte count, total blood cholesterol level is included to calculate the CONUT score. These indices are reliable prognostic biomarkers in patients with cancer [13,14,15,16,17] or cardiovascular disease [18]. PNI may be used to assess mortality risk in patients with sepsis [19,20]; however, its prognostic efficacy in sepsis, as with GNRI and CONUT, remains unclear due to limited evidence [21,22].

Currently, we know of no published studies that compare the utility of these three nutritional indices in patients with sepsis. Therefore, the purpose of this study was to investigate the effect of the GNRI, PNI, and CONUT score on mortality in patients with sepsis by analysing survival curves for each index using a large, nationwide registry database.

## 2. Methods

### 2.1. Study Setting and Data Source

The electronic medical records from which the data used in this retrospective observational study were obtained were provided by Medical Data Vision (MDV, Tokyo, Japan). The MDV database contains data from over 400 acute-care hospitals that include anonymized electronic health insurance claims and diagnosis procedure combinations accounting for about 23% of all claims made in Japan. Therefore, this large-scale database contains data on approximately 30 million patients. The data extracted included information on age, sex, laboratory values, admission date, primary diagnoses, concomitant diagnoses, complication diagnoses, medical procedures, prescriptions, drug administration, discharge status, and hospital length of stay. The diagnoses were recorded based on the International Classification of Diseases Tenth Revision (ICD-10) codes. The patient data used in the present study were obtained from 42 acute-care hospitals with laboratory data among all acute-care hospitals registered in the MDV database. The study period was between January 2011 and December 2019.

This study was conducted in accordance with the principles of the Declaration of Helsinki. The study protocol was approved by the Institutional Review Board of Osaka General Medical Center, Osaka, Japan (approval no. #S201916015). Due to the pre-existing and anonymized data stored in an un-linkable manner, the requirement for informed consent was waived.

### 2.2. Study Population

The criteria for patient inclusion in this study were age > 16 years, diagnosis of infectious disease, diagnosis of sepsis, and the need for an unplanned hospital admission between January 2011 and December 2019. In this study, infectious disease was defined by including any of the ICD-10 infection codes previously proposed by the Institute for Health Metrics and Evaluation (IHME) [23] in the primary diagnosis or the diagnosis on admission. Sepsis-3 was defined as an increase in the retrospectively calculated total Sequential Organ Failure Assessment (SOFA) score of ≥2 points on admission. Patients meeting the following exclusion criteria were excluded: diagnosis other than sepsis (SOFA score < 2) and incomplete clinical and date data.

### 2.3. Data Collection

For evaluation of baseline patient characteristics, we collected the following data: age, sex, date of admission, Charlson comorbidity index (CCI) [24], SOFA score and SOFA sub-scores, ICU admission, catecholamine use, surgery with general anaesthesia, and underlying Sepsis-3 and disseminated intravascular coagulation values. Data on patient clinical characteristics, demographics, laboratory test results, and comorbidities were collected from the patients’ medical records. To calculate PNI, GNRI, and the CONUT score, we recorded serum albumin, total cholesterol, C-reactive protein, and total peripheral lymphocyte counts. The presence of malnutrition at hospital admission was defined according to these three nutritional indices, as shown in Table 1.

### 2.4. Statistical Analysis

Descriptive statistics are summarized as group medians with the interquartile range for continuous variables and as frequencies with percentages for categorical variables. The Mann–Whitney U test or Chi-square test was used to compare baseline characteristics between the survivors and non-survivors.

We evaluated the non-linear associations between mortality and nutrition indices. We also used logistic regression to fit restricted cubic spline models. Reference points were determined based on each parameter’s normal value: 400 mg/dL for GNRI, 150 × 10^3^/μL for PNI, and 1.0 for CONUT. Knot values, which were placed at equally spaced percentiles of the original variable’s marginal distribution, were established on the basis of Harrell’s recommended percentiles [25]. The Wald test was used to determine the number of knots in each analysis such that the explanatory variables at all sections divided by the knots were significant [26]. Then, to evaluate the significance of these three nutritional indices on mortality in a time-dependent manner, we constructed Kaplan–Meier survival curves by the specific cut-off values of these indices.

All hypotheses were two-sided. A *p*-value of <0.05 was considered to indicate statistical significance. Cases with missing data in the regression models were excluded from the analyses. All statistical analyses were conducted using STATA Data Analysis and Statistical Software version 14.0 (StataCorp, College Station, TX, USA) and JMP software version 15.0 (SAS Institute, Tokyo, Japan).

## 3. Results

### 3.1. Patient Eligibility Outline

A flowchart outlining the patients eligible for inclusion in this study is shown in Figure 1. The total number of inpatients with an infectious disease during the study period was 171,596. Following the application of the inclusion and exclusion criteria, 32,159 patients with sepsis remained for inclusion in the present study.

### 3.2. Baseline Characteristics

The baseline patient characteristics according to each diagnosis are shown in Table 2. The median age, BMI, and CCI of the patients were 79 years, 21.7 kg/m^2^, and 3, respectively. The non-survivors were significantly older than the survivors (84 vs. 79 years, *p* < 0.001). The median BMI was significantly lower in non-survivors than in survivors (19.9 vs. 21.8 kg/m^2^, *p* < 0.001). There was a significant variation in the source of infection between the two groups (*p* < 0.001). Laboratory tests showed that the median level of albumin (2.8 vs. 3.4 g/dL, *p* < 0.001) and lymphocyte count (699 vs. 855/μL, *p* < 0.001) were significantly lower in the non-survivors versus survivors, as were the median GNRI (80.0 vs. 92.8, *p* < 0.001) and PNI (32.1 vs. 38.9, *p* < 0.001). The median CONUT score was significantly higher in the non-survivors versus survivors (7 vs. 4, *p* < 0.001).

### 3.3. Mortality

Restricted cubic splines were performed in the multivariate logistic models to deeply assess any non-linear association between each nutrition index and mortality. Although a GNRI within a range from 100 to 200 indicated no remarkable change in predicted mortality, when the GNRI dropped below 100, the risk of mortality rose sharply (Figure 2A). Similarly, the risk of mortality rose sharply as PNI dropped below approximately 40 (Figure 2B). For both GNRI and PNI, the shapes of the non-linear cubic spline curves between the two groups were similar. An increase in the CONUT score was associated with an increase in mortality in an apparent linear manner (Figure 2C).

Furthermore, the association between survival rate and each nutrition index is shown in Figure 3. Both lower levels of GNRI and PNI and higher levels of CONUT score were associated with lower survival rates.

### 3.4. Subgroup Analysis

We performed subgroup analysis for the ICU admission group (*n* = 1804) and non-ICU admission group (*n* = 30,355). Restricted cubic splines were performed in the multivariate logistic models to assess any non-linear association between each nutrition index and mortality for each group (Appendix A). The results for each of the nutritional indices in the non-ICU group were similar to those for the overall population (Appendix A). In the ICU group, however, although the shapes of the curves were also similar to those for the overall population, the confidence interval of estimated mortality was wide due to the small number of patients.

## 4. Discussion

This study used a large cohort of patients with sepsis in Japan to investigate a potential association between three nutrition indices and mortality. The mortality risks rose sharply as levels of PNI and GNRI decreased below approximately 40 and 100, respectively. We conducted the present study as a revalidation of the clinical significance of the PNI and GNRI and to determine cut-off values, not to compare these markers with each other. Our findings may increase the clinical value of the GNRI and PNI through the use of these markers in clinical settings to aid in decision-making. Overall, the findings in the present study appear to suggest that the GNRI, PNI, and CONUT may be meaningful indicators for determining survival in patients with sepsis.

These nutritional indices have been well-studied in other fields so far. Expression of the PNI and tumour-infiltrating lymphocytes (TILs) score was associated with clinical outcomes in esophageal cancer, which would support their roles as prognostic biomarkers [13]. The relationship between PNI and TILs indicates that nutritional status and systemic immune competence may influence patient prognosis via a local immune response. In their recent retrospective, single-centre study, Oba et al. [27] showed lower PNI to be significantly associated with lower disease-free survival in patients with breast cancer who underwent neoadjuvant chemotherapy. In addition, another recent retrospective study reported the association of PNI with an increase in the rate of in-hospital mortality and independent predictors of mortality in patients with infective endocarditis [18].

Otherwise, other than in cancer research, there is still limited research evaluating the associations of these nutritional indices with survival from sepsis. In this study, we analysed a large amount of real-world data. Our findings showed that lower GNRI and PNI scores were associated with a lower survival rate, as were higher CONUT scores. Previously, Shimoyama et al. reported PNI in patients with sepsis to be a predictor of both increased mortality [28] and of septic acute kidney injury and an indicator for the initiation of renal replacement therapy [29]. A prospective cohort study reported that increasing the amount of albumin improves the prognosis of patients with severe sepsis [30]. From Table 1, GNRI and PNI show a positive correlation with the level of albumin, meaning that GNRI and PNI values may potentially be helpful for predicting prognostic in sepsis [31].

Our study has several limitations. First, there may have been errors in the diagnosis of sepsis recorded in the MDV database because the accuracy of diagnoses recorded in such administrative claims databases is generally lower than that of diagnoses recorded in prospective studies. Similarly, the MDV database may have included under- or over-estimation and misclassification of the underlying conditions at data entry. Second, the results may have been influenced by a large amount of missing data for PNI and CONUT scores. Third, the patient numbers in the ICU group comprised only about 5% of all patients. This might have affected the shape of the non-linear cubic spline curves of the nutrition indices against mortality and reduced statistical power in the ICU group.

## 5. Conclusions

In conclusion, through the use of a multi-centre cohort study database in Japan, the present study showed a non-linear association between both PNI and GNRI values and mortality in patients with sepsis. Our findings might suggest the potential value of the GNRI and PNI in sepsis management to identify patients who may be at high risk for death.

## Figures and Tables

**Figure 1 diagnostics-13-01302-f001:**
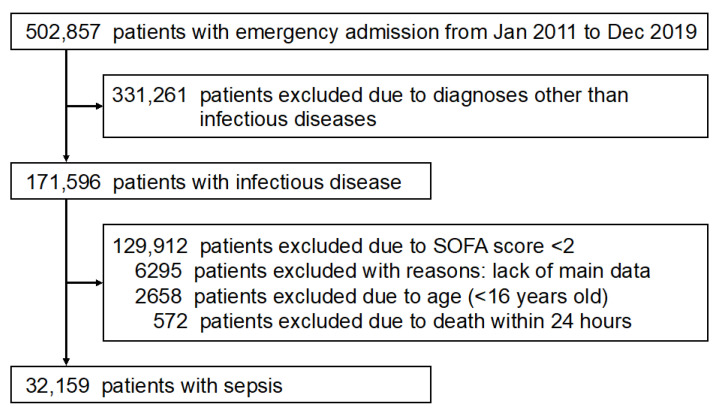
Patient flow diagram. SOFA, Sequential Organ Failure Assessment.

**Figure 2 diagnostics-13-01302-f002:**
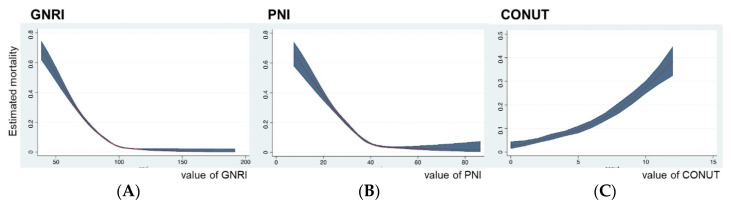
Non-linear cubic spline curve of nutrition indices against mortality in sepsis. (**A**), GNRI (Geriatric Nutritional Risk Index). (**B**), PNI (Prognostic Nutritional Index). (**C**), CONUT (Controlling Nutrition Status).

**Figure 3 diagnostics-13-01302-f003:**
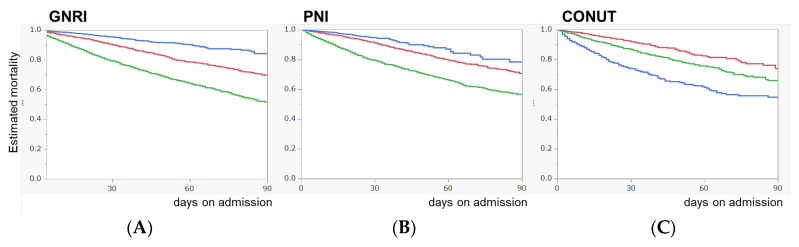
Survival curve of nutrition indices against mortality in sepsis. (**A**), GNRI (Geriatric Nutritional Risk Index). (**B**), PNI (Prognostic Nutritional Index). (**C**), CONUT (Controlling Nutrition Status).

**Table 1 diagnostics-13-01302-t001:** Definition of nutrition indices used in this study.

Nutritional Indices	Normal	Mild	Moderate	Severe
GNRI (Geriatric Nutritional Risk Index)				
14.89 × albumin (g/dL) + 41.7 × body weight/ideal body weight	>98	92–98	82–91	<82
PNI (Prognostic Nutritional Index)				
10 × albumin (g/dL) + 0.005 × total lymphocyte count (mm^3^)	>38	–	35–38	<35
CONUT (Controlling Nutrition Status) score				
Albumin, g/dL (score)	≥3.5 (0)	3.0–3.4 (2)	2.5–2.9 (4)	<2.5 (6)
Cholesterol, mmol/L (score)	>4.65 (0)	3.62–4.65 (1)	2.59–3.61 (2)	<2.59 (3)
Total lymphocyte count, ×10^9^/L (score)	≥1.60 (0)	1.20–1.59 (1)	0.80–1.19 (2)	<0.8
Overall score	0–1	2–4	5–8	9–12

**Table 2 diagnostics-13-01302-t002:** Patient characteristics.

	Total	Survivors	Non-Survivors	*p*-Value	Missing
	(*n* = 32,159)	(*n* = 28,698)	(*n* = 3461)		
Age, years, median (IQR)	79 (69–86)	79 (68–86)	84 (76–89)	<0.001	0
Male sex, *n* (%)	19,069 (59.3%)	17,027 (59.3%)	2042 (59.0%)	0.708	0
Body mass index, median (IQR)	21.7 (19.1–24.4)	21.8 (19.3–24.6)	19.9 (17.5–22.9)	<0.001	3803 (11.8%)
Charlson comorbidity index, median (IQR)	5 (2–9)	5 (2–9)	4 (2–8)	0.113	0
Total SOFA score, median (IQR)	3 (2–4)	3 (2–4)	4 (2–5)	<0.001	0
ICU admission, *n* (%)	1804 (5.6%)	1394 (4.9%)	410 (11.8%)	<0.001	0
Disseminated intravascular coagulation, *n* (%)	2467 (7.7%)	1880 (6.6%)	587 (17.0%)	<0.001	0
Catecholamine use, *n* (%)	1711 (5.3%)	1170 (4.1%)	541 (15.6%)	<0.001	0
Renal replacement therapy, *n* (%)	956 (3.0%)	699 (2.4%)	257 (7.4%)	<0.001	0
Source of infection, *n* (%)				<0.001	0
Respiratory	13,409 (41.7%)	11,346 (39.5%)	2063 (59.6%)		
Abdominal	7436 (23.1%)	6839 (23.8%)	597 (17.2%)		
Urinary tract	5444 (16.9%)	5099 (17.8%)	345 (10.0%)		
Bone/soft tissue	1435 (4.5%)	1347 (4.7%)	88 (2.5%)		
Central nervous system	452 (1.4%)	406 (1.4%)	46 (1.3%)		
Cardiovascular	342 (1.1%)	306 (1.1%)	36 (1.0%)		
Other	3641 (11.3%)	3355 (11.7%)	286 (8.3%)		
Laboratory data, median (IQR)					
Total protein, g/dL	6.6 (6.1–7.1)	6.7 (6.2–7.2)	6.2 (5.6–6.8)	<0.001	2112 (6.6%)
Albumin, g/dL	3.4 (2.9–3.8)	3.4 (3.0–3.8)	2.8 (2.3–3.3)	<0.001	0
Bilirubin, mg/dL	1.0 (0.6–1.6)	1.0 (0.6–1.7)	0.8 (0.5–1.4)	<0.001	874 (2.7%)
Creatinine, mg/dL	1.1 (0.7–1.7)	1.0 (0.7–1.6)	1.2 (0.8–2.0)	<0.001	42 (0.1%)
Blood urea nitrogen, mg/dL	23.0 (15.8–36.0)	22.0 (15.3–34.1)	31.9 (21–49.2)	<0.001	1622 (5.0%)
Total cholesterol, mg/dL	152 (126–181)	154 (128–182)	139 (110–170)	<0.001	23,248 (72.3%)
Glucose, mg/dL	130 (109–166)	130 (109–165)	132 (106–174)	0.472	11,555 (35.9%)
C-reactive protein, mg/dL	8.1 (2.7–15.9)	7.9 (2.6–15.7)	10.0 (4.3–18.0)	<0.001	639 (2.0%)
White blood cell count, /µL	9950 (6890–13,850)	9920 (6900–13,800)	10,030 (6650–14,500)	0.233	37 (0.1%)
Lymphocyte count, /µL	838 (534–1273)	855 (547–1292)	699 (435–1098)	<0.001	13,975 (43.5%)
Red blood cell count, ×10^4^/µL	392 (339–444)	396 (343–447)	357 (302–412)	<0.001	36 (0.1%)
Haemoglobin, g/dL	12.1 (10.5–13.7)	12.2 (10.6–13.8)	11.0 (9.4–12.6)	<0.001	36 (0.1%)
Platelet count, 10^4^/µL	15.9 (11.9–22.0)	15.9 (12.0–21.9)	15.9 (10.8–23.1)	0.104	35 (0.1%)
PNI, median (IQR)	38.3 (33.0–43.4)	38.9 (33.9–43.9)	32.1 (27.0–37.6)	<0.001	13,975 (43.5%)
GNRI, median (IQR)	91.6 (81.9–101.0)	92.8 (83.4–101.9)	80.0 (71.4–89.1)	<0.001	3803 (11.8%)
CONUT, median (IQR)	5 (3–7)	4 (3–7)	7 (4.5–9)	<0.001	26,313 (81.8%)

Data are expressed as percent or median with interquartile range (IQR), as indicated. IQR, interquartile range; SOFA, Sequential Organ Failure Assessment; ICU, intensive care unit; PNI, Prognostic Nutritional Index; GNRI, Geriatric Nutritional Risk Index; CONUT, Controlling Nutrition Status.

## Data Availability

The data of this report are available from the corresponding authors upon request.

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
