# Peer review of "Comparison of Nutrition Indices for Prognostic Utility in Patients with Sepsis: A Real-World Observational Study"

_diagnostics, 2023, doi:10.3390/diagnostics13071302_

Round 1

Reviewer 1 Report

Thank you very much for allowing me to read this interesting article. The Comparison of nutrition indices for prognostic utility in patients with sepsis: a real-world observational study

Wriiten by Django Kyo 1 , Shiho Tokuoka 1, Shunsuke Katano 1, Ryo Hisamune and et al

 It is  a retrospective study  look into a large database of  42 acute care hospital on  the impact of three nutritional indices,  with the use of Geriatric Nutritional Risk Index (GNRI), Prognostic Nutritional Index (PNI), and Controlling Nutritional Status (CONUT), on mortality in patients with sepsis in Japan

They  identified 32,159 patients with sepsis according to the inclusion criteria.

Of them, 1,804 patients were treated in intensive care units,

and 3,461 patients were non-survivors.  3461 were mortality,

Does it mean most of them expired outside the ICU??

If so, The management in and out of the ICU should be different so the the mortality should be effected by this. So the author should clarify this before make a firm conclusion. 

Patients in and out of ICU should be analysed  seperately or make a limitation of analysis.

Author Response

We appreciate you taking the time to review our manuscript. Thank you very much for your valuable comment. As your point put, it was true that almost all patients haven’t admitted to ICU. Therefore, we have separately analyzed any nonlinear association between each nutrition index and mortality in ICU admitted group and non-ICU admitted group. As these results, we have added the sentence and reference. We have added the sentences, supplementary figures, and reference as you suggested, as follows:

“3.4       Subgroup analysis

We performed subgroup analysis for the ICU admission group (n=1,804) and non-ICU admission group (n=30,355). Restricted cubic splines were performed in the multivariate logistic models to assess any nonlinear association between each nutrition index and mortality for each group (Supple. Fig. 1). The results for each of the nutritional indices in the non-ICU group were similar to the those for the overall population (Supple. Fig. 1A- C). In the ICU group, however, although the shapes of the curves were also similar to those for the overall population, the confidence interval of estimated mortality was wide due to the small number of patients.” (Page 6, line 166-173)

“Third, the patient numbers in the ICU group comprised only about 5% of all patients. This might have affected the shape of the non-linear cubic spline curves of the nutrition indices against mortality and reduced statistical power in the ICU group.” (Page 7, line 211-213)

“Supplement Figures” (new file)

“1. Egi M, Ogura H, Yatabe T, Atagi K, Inoue S, et al. The Japanese Clinical Practice Guideline for Management of Sepsis and Septic Shock 2020 (J-SSCG 2020). Journal of Intensive Care 2021, 9, 53. doi: 10.1002/ams2.659.

2. Evans L, Rhodes A, Alhazzani W, Antonelli M, Coopersmith CM, et al. Survivng sepsis campaign: international guideline for management of sepsis and septic shock 2021. Intensive Care Med 2021, 47(11), 1181-1247. doi: 10.1007/s00134-021-06506-y.

3. De Waele E, Malbrain M, Spapen H. Nutrition in sepsis: a bench-to-bedside review. Nutrients 2020, 12(2). doi: 10.3390/nu12020395.

7. Felblinger DM. Malnutrition, infection, and sepsis in acute and chronic illness. Crit Care Nurs Clin North Am 2003, 15(1), 71-78. doi: 10.1016/s0899-5885(02)00040-0.

9. Bourke CD, Berkley JA, Prendergast AJ. Immune dysfunction as a cause and consequence of malnutrition. Trends Immunol 2016, 37(6), 386-398. doi: 10.1016/j.it.2016.04.003.

14. Mohri Y, Inoue Y, Tanaka K, Hiro J, Uchida K, et al. Prognostic nutritional index predicts postoperative outcome in colorectal cancer. World J Surg 2013, 37(11), 2688-2692. doi: 10.1007/s00268-013-2156-9.

15. Mori S, Usami N, Fukumoto K, Mizuno T, Kuroda H, et al. The significant of the prognostic nutritional index in patients with completely resected non-small cell lung cancer. PLoS One 2015, 10(9), e0136897. doi: 10.1371/journal.pone.0136897.

16. Caputo F, Dadduzio V, Tovoli F, Bertolini G, Cabibbo G, et al. The role of PNI to predict survival in advanced hepatocellular carcinoma treated with Sorafenib. PLoS One 2020, 15(5), e0232449. doi: 10.1371/journal.pone.0232449.

17. Huang X, Hu H, Zhang W, ShaoZ. Prognostic value of prognostic nutritional index and systemic immune-inflammation index in patients with osteosarcoma. J cell Physiol 2019, 234(10), 18408-18414. doi: 10.1002/jcp.28476.

20. Li T, Qi M, Dong G, Li X, Xu Z, et al. Clinical value of prognostic nutritional index in prediction of the presence and severity of neonatal sepsis. Journal of Inflammation Research 2021, 21, 7181-7190. doi: 10.2147/JIR.S343992.

21. Lee JS, Choi HS, Ko YG, Yun DH. Performance of the Geriatric Nutritional Risk Index in predicting 28-day hospital mortality in older adult patients with sepsis. Clin Nutr 2013, 32(5), 843-848. doi: 10.1016/j.clnu.2013.01.007.

22. Yamaguchi J, Kinoshita K, Nakagawa K, Mizuochi M. Undernutrition Scored Using the CONUT Score with Hypoglycemic Status in ICU-Admitted Elderly Patients with Sepsis Shows Increased ICU Mortality. Diagnostics 2023, 13(4), 762. doi: 10.3390/diagnostics13040762.

Reviewer 2 Report

Reviewer Comments to authors

Regarding the study entitled "Comparison of nutrition indices for prognostic utility in patients with sepsis: a real-world observational study" with accept.

It is a good research work. The paper is written in good English.

Abstract: Well summarized.

Introduction: Methods and interventions are described in sufficient details.

Methodology: Enough information is provided.

Results: Tables and figure are well selected and related to the main research question.

Discussion: Discussion provides an answer to the research question

Conclusion: Findings are well summarized

References: References are up to date and can be easily accessed.

Figure and Tables: They are selected properly and related to the main research question.

Author Response

We appreciate you taking the time to review our manuscript. Thank you for your comments.